# Practical No-box Adversarial Attacks against DNNs

**Qizhang Li** [*]
ByteDance AI Lab
liqizhang@bytedance.com

**Yiwen Guo** [†]
ByteDance AI Lab
guoyiwen.ai@bytedance.com

**Hao Chen**
University of California, Davis
chen@ucdavis.edu

## Abstract

The study of adversarial vulnerabilities of deep neural networks (DNNs) has progressed rapidly. Existing attacks require either internal access (to the architecture, parameters, or training set of the victim model) or external access (to query the model). However, both the access may be infeasible or expensive in many scenarios. We investigate *no-box* adversarial examples, where the attacker can neither access the model information or the training set nor query the model. Instead, the attacker can only gather a small number of examples from the same problem domain as that of the victim model. Such a stronger threat model greatly expands the applicability of adversarial attacks. We propose three mechanisms for training with a very small dataset (on the order of tens of examples) and find that prototypical reconstruction is the most effective. Our experiments show that adversarial examples crafted on prototypical auto-encoding models transfer well to a variety of image classification and face verification models. On a commercial celebrity recognition system held by `clarifai.com`, our approach significantly diminishes the average prediction accuracy of the system to only $15.40\%$, which is on par with the attack that transfers adversarial examples from a pre-trained Arcface model. Our code is publicly available at: `https://github.com/qizhangli/nobox-attacks`.

## 1 Introduction

The adversarial vulnerability of deep neural networks (DNNs) have been extensively studied over the past few years [46, 12, 35, 30, 3, 29, 1]. It has been demonstrated that an attacker is able to generate small, human-imperceptible perturbations to fool advanced DNN models to make incorrect decisions. These attacks pose great threats to security-critical systems where DNNs are deployed and lead to increasing concerns about the robustness of DNNs.

Based on how much information the attacker knows, we can divide existing attacks into white-box and black-box settings. In black-box attacks, the attacker cannot access the architecture, the parameters, or the training data of the victim model. Early attempts for black-box adversarial attacks [33, 34] relied on the transferability of adversarial examples. They trained substitute architectures by querying the victim models. Recent progress considered gradient estimation [5, 20, 50, 21, 13, 6, 4] and boundary tracing [2].

Black-box attacks rely on querying the victim models, *a.k.a.*, "oracles". However, in many scenarios, such queries are either infeasible (e.g., the model API is inaccessible to the attacker) or are expensive in time or money. To overcome this limitation, we consider a stronger threat model where the attacker makes *no query* to the victim model (and also has no access to the model parameters or its training data). This was coined as the "no-box" threat model [5] but no practical attack has been studied to the best of our knowledge. We investigate such no-box attacks against DNNs on computer vision models. Similar to some strong black-box attacks [34], we assume that the attacker can access neither a large

---

[*]Work done during an internship at ByteDance
[†]Corresponding author

scale training data nor pre-trained models on it. Instead, she or he can collect only a small number of auxiliary examples (on the order of tens or less) from the same problem domain. Given this small sample size, it is challenging to obtain a substitute model using conventional supervised training.

Inspired by recent advances in unsupervised representation learning and distribution modeling [48, 41], we developed *auto-encoders* that can learn discriminative features *given very little data*, *e.g.*, 20 images from 2 classes. We investigated three training mechanisms by (a) estimating the front view of each rotated image, (b) estimating the best fit of each possible jigsaw puzzle, and (c) constructing prototypical images, respectively. They entail discriminative and, more importantly, generalizable feature representations. We evaluated our approach on two computer vision tasks: image classification and face verification. Our experiments show that adversarial examples crafted on such auto-encoding models transfer well to a variety of open-source victim models, and their effectiveness is sometimes even on par with those crafted using pre-trained models trained on the same large-scale data set as the victim models. On a celebrity recognition system hosted by `clarifai.com`, our approach reduced the prediction accuracy of the system from $100.00\%$ to only $15.40\%$, using only 10 facial images for training and crafting each adversarial example. We also studied the quality of the generated adversarial example in such a way, and we showed in the supplementary materials that the generated no-box adversarial examples were intrinsically different from the adversarial examples generated in the white-box/black-box settings, which probably worth further exploring.

## 2 Related Work

**Adversarial attacks**    The common goal of adversarial attacks is to generate small perturbations that is capable of fooling learning machines [46, 12]. Current adversarial attacks can be roughly divided into two categories: white-box attacks and black-box attacks, according to whether the training data, architecture, and parameters of the victim model are accessible [33]. Black-box attacks target the scenario where very limited information about the victim model is accessible, while a certain number of model queries (*a.k.a.*, oracle queries) are granted. This line of work can further be split into two sub-lines, revolving around "reverse engineering" via training substitute models [49, 33, 34, 60] and gradient/boundary estimations [31, 5, 9, 54, 2, 4], respectively, both of which have their pros and cons. In particular, they all require a large number of timely queries to the victim model, for different purpose though. However, it is also conceivable that numerous queries and real-time response from the victim model are difficult to obtain, if not infeasible. In this paper we consider a critical threat by performing adversarial attacks to models, on which no query is to be issued. It was mentioned once as a "no-box" setting [5] yet has not been studied in depth. Our work is also related to attacks to auto-encoders [25, 47] and image translation networks [37], despite they considered different tasks.

**Transferability**    Our approach exploits the transferability of adversarial examples. It was first unraveled by Szegedy *et al.* [46] that adversarial examples crafted on one DNN may fool (*i.e.*, transfer to) other DNNs with a non-trivial success rate. Single-step attacks, *e.g.*, the fast gradient sign method (FGSM) [12], and multi-step attacks, *e.g.*, the iterative FGSM (I-FGSM) [26], have been compared in the sense of transferability [26]. Multiple efforts have been devoted to improving the transferability of adversarial examples. For instance, Xie *et al.* [53] suggested to apply random and differentiable transformations to the inputs when performing attacks on pre-trained substitute models [53]. It was also widely explored to craft more transferable adversarial examples by optimizing on intermediate layers of the substitute models [61, 22, 19, 27, 14]. Unlike prior work that experimented on substitute models trained on the same set of data as that for training victim models, in this paper, we assume no access to the real victim training set and attempt to obtain models on a small number of auxiliary examples, *e.g.*, 20 images from 2 classes. Moreover, in lieu of generating adversarial examples on softmax-based classification networks, we resort to auto-encoding models and (possibly for the first time) attempt to develop suitable attacks for them.

**Self-supervised learning**    Self-supervised learning exploits surrogate supervision from unlabeled data for gaining high-level data understanding of data, targeting a similar problem to ours. Pretext learning tasks including context prediction [8], image colorization [59], rotation prediction [11], and jigsaw puzzle analysis [32] have been introduced over the last few years. Our *reconstruction from chaos* mechanisms are inspired by self-supervised learning. Though inferior to our best ones from *prototypical image reconstruction*, they achieve reasonable performance in transferring adversarial examples to the supervised victim models. More discussions will be given in Section 3, 4.2, and 4.3.

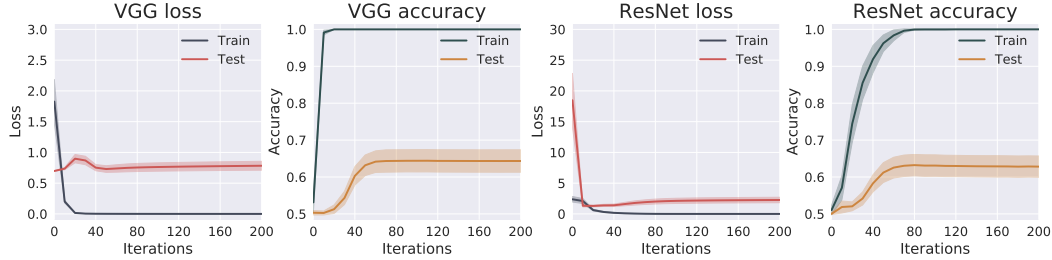

Figure 1: With limited training data, conventional supervised learning suffer from severe over-fitting.

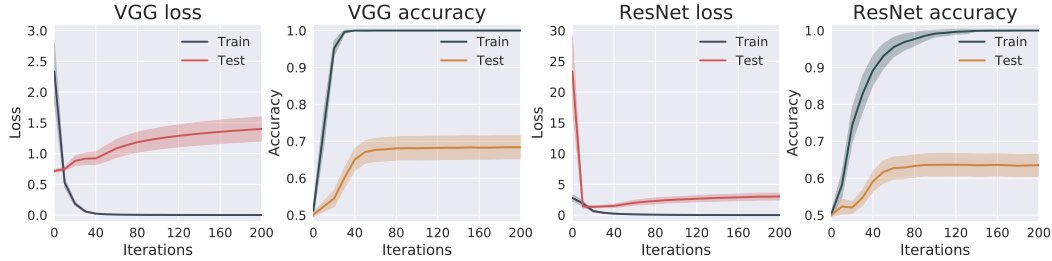

Figure 2: Data augmentations and regularizations help to a limited extent in the conventional supervised setting. Weight decay, dropout, and some popular data augmentations are adopted.

## 3 Our Approach to No-Box Attacks

Our work considers a threat model in which the attacker attempts to attack classification models without issuing any model query. We consider the practical scenario where the attacker can gather a (labeled) dataset with very limited size, but not a large-scale training dataset or pre-trained models on it. What comes uppermost in mind is to utilize the transferability of adversarial examples. However, current supervised learning for DNNs require large-scale training to generalize, hence, to achieve the goal of performing attack under our threat model, one should first develop proper training mechanisms and "substitute" architectures. We will introduce our design in Section 3.1. Given the substitute models, one should then perform proper attacks with outstanding transferability, which will be elaborated in Section 3.2.

### 3.1 Training Mechanisms

Assume a benign instance $x_0$ is to be perturbed such that being mis-classified into an arbitrary label. We aim to train a discriminative model on a small and thus easily gathered (and labeled) auxiliary dataset $\mathcal{X} := \{(x_i, y_i)\}_{i=0}^{n-1}$, including the instance $x_0$ to be perturbed. We first consider the auxiliary dataset involving only two classes, *i.e.*, $y_i \in \{0, 1\}$, despite higher data variety is always beneficial. Throughout this paper, we constrain $n \leq 20$ if not otherwise clarified.

Since the sample size is limited, we ought to make best use of data. It is conceivably very challenging to train a classification DNN equipped with softmax via *conventional supervised learning*, using the cross-entropy loss. One may train a variety of DNNs on $\mathcal{X}$ and observe fast convergence. Yet, test performance will show little generalizable discriminative ability of all these models, on account of over-fitting (as depicted in Figure 1, shaded areas represent scaled standard deviations). Training on different data from that of the victim models also leads to low attack success rates, implying that the internal statistics of images are not really captured. Data augmentations and typical regularizations (*e.g.*, dropout [43] and weight decay) provide limited help in improving the models, showing that these methods does not play fundamental roles in generalization [57] (see Figure 2).

One may resort to image-to-image mappings, owing to their success in modeling internal distribution of data, even using single images [48, 41]. A natural baseline in this spirit is a classical convolutional auto-encoder learning to reconstruct its inputs, *i.e.*, minimizing $\sum_i \|\text{Dec}(\text{Enc}(x_i)) - x_i\|^2$. On one hand, such a model is capable of capturing low-level image representations without suffering from severe over-fitting, while on the other hand, discriminative ability is by no means entailed and thus adversarial examples crafted against the model are difficult to transfer to the victim models, as will be empirically discussed in Section 4.2 and 4.3.

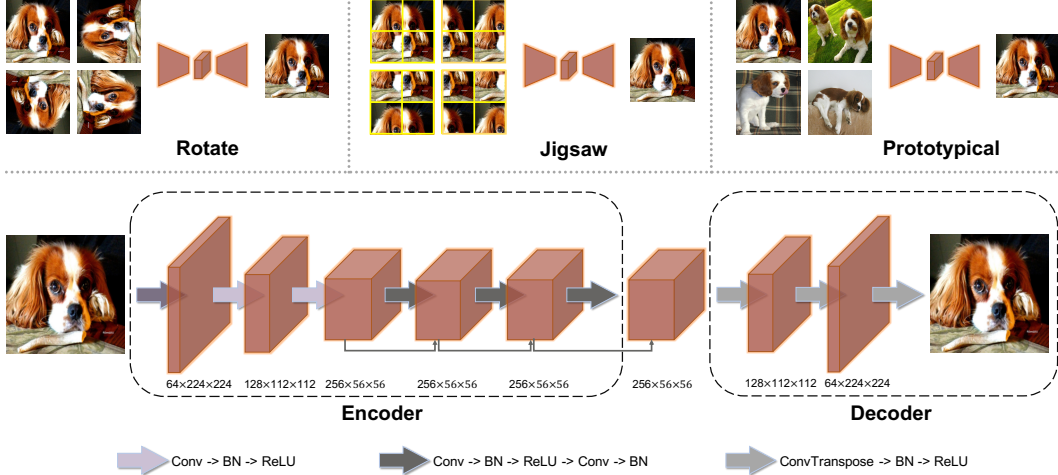

Figure 3: Illustration of the proposed training mechanisms for auto-encoding substitute models for no-box attacks, including two unsupervised mechanisms (*i.e.*, reconstruction from rotation/jigsaw) and a supervised mechanism (*i.e.*, prototypical image reconstruction).

**Reconstruction from chaos**     Less over-fitting implies more "resonation" between DNN architecture and (joint) data distribution [51]. We thus stick with image-to-image models and try to enhance their discriminative ability. Our initial attempts are inspired by *self-supervised learning* [10, 32, 11]. It has been shown that predicting image rotation angles [11] and jigsaw puzzle configurations [32] endow DNNs with considerable discriminative abilities. We thus incorporate such pretext information and introduce tasks for estimating: (a) the front view of each rotated images and (b) the prefect fit of each possible jigsaw puzzle, using image-to-image auto-encoding models. Their learning objectives are commonly formulated as:

$$L_{\text{rotation/jigsaw}} = \frac{1}{n} \sum_{i=0}^{n-1} \|\text{Dec}(\text{Enc}(T(x_i))) - x_i\|^2, \tag{1}$$

in which the image transformation function $T(\cdot)$ is designed to rotate images or shuffle image patches for the two tasks, respectively. Though the learning task is still *unsupervised* as with the classical auto-encoder, the pretext information is content-related and should be beneficial to classification. We will show in Section 4.2 that adversarial examples crafted on substitute models trained using these mechanisms transfer better than the classical unsupervised baseline.

**Prototypical image reconstruction**     Besides reconstruction from "chaos" (*i.e.*, rotations and jigsaw puzzles), a more supervised mechanism is also introduced. Since the attacker also has $\{y_i\}$ at hand, there is no reason not to use it. In this context, we propose to encourage the models to reconstruct class-specific prototypes, such that direct supervision is entailed to the auto-encoding models. Natural images in $\{x_i\}$ are eligible to be such prototypes, making the model outputs lie in pixel spaces as well. More formally, we opt to minimize

$$L_{\text{prototypical}} = \frac{1}{n} \sum_{i=0}^{n-1} \left( (1 - y_i) \left\| \text{Dec}(\text{Enc}(x_i)) - x^{(0)} \right\|^2 + y_i \left\| \text{Dec}(\text{Enc}(x_i)) - x^{(1)} \right\|^2 \right), \tag{2}$$

in which $x^{(0)} \in \{x_i | y_i = 0\}$ and $x^{(1)} \in \{x_i | y_i = 1\}$ are randomly chosen image prototypes from the two classes, respectively. The intuition behind this mechanism is that a model will have to distinguish samples with different labels, in order to obtain perfect training. See Figure 3 for an outline of this prototypical reconstruction mechanism, together with those of the other two mechanisms introduced in the previous paragraph. It is possible to introduce more than one decoder with this mechanism, by sampling multiple pairs of image prototypes from the two classes. The loss function in Eq. (2) can also be easily generalized to train such models with multiple decoders. Since richer supervision can be obtained from more decoders and more prototypes, we may expect superior attack performance in such a setting. Benefit from explicit supervision, the obtained prototypical models may also be cast into classification models. We achieve the goal by making predictions based on similarity between

model outputs and the prototypes from different classes. That said, an input instance $x$ is predicted into a class from which the similarity between its chosen prototype $x^{(y)}$ and the reconstruction output $\text{Dec}(\text{Enc}(x))$ is maximized. If the Euclidean distance is utilized for measuring similarity, then

$$\hat{y} = \underset{y \in \{0,1\}}{\arg \min} \left\| \text{Dec}(\text{Enc}(x)) - x^{(y)} \right\|. \tag{3}$$

The obtained models are capable of making predictions with many informative features retained in the code representing inputs. In contrast to the reconstruction from chaos mechanisms that are still unsupervised, the prototypical reconstruction mechanism is supervised. We will show in Section 4.2 and 4.3 that the prototypical reconstruction models suffer less from over-fitting than the conventional supervised baseline, and adversarial examples craft on them transfers significantly better.

## 3.2 Attacks with Improved Transferability

It is non-trivial to adopt existing gradient-based attacks (*e.g.*, FGSM [12], I-FGSM [26], PGD [29], *etc*) for crafting adversarial examples on auto-encoders. Hence, before going into experiments, we first explain how attacks are performed on the image-to-image auto-encoding models. We get started by introducing an adversarial loss for the models, that are trained to reconstruct $\tilde{x}_i$ for each input $x_i$:

$$L_{\text{adversarial}} = -\log p(y_i|x_i) \quad \text{where} \quad p(y_i|x_i) = \frac{\exp\left(-\lambda \|\text{Dec}(\text{Enc}(x_i)) - \tilde{x}_i\|^2\right)}{\sum_j \exp\left(-\lambda \|\text{Dec}(\text{Enc}(x_i)) - \tilde{x}_j\|^2\right)}, \tag{4}$$

in which $\lambda > 0$ is a scaling parameter. Specifically, we have $\tilde{x}_i := x_i$ for reconstructing images from rotations and jigsaw puzzles, and we have $\tilde{x}_i := x^{(0)}$ for the prototypical reconstruction models with $x_i$ labeling as $y_i = 0$. Such an $\tilde{x}_i$ is called the positive prototype for $x_i$, and their distance is enlarged while the adversarial loss $L_{\text{adversarial}}$ is maximized by attacks. For $\tilde{x}_j$s that act as negative prototypes, we sample them from the other class. $L_{\text{adversarial}}$ is a cross-entropy-flavored loss that involves both the positive and negative prototypes, and many gradient-based baseline attacks (*e.g.*, FGSM [12], I-FGSM [26], PGD [29], *etc*) can be readily adopted based on it. In this paper, we perform attacks in conjunction with ILA [19], to achieve more powerful threats from the auto-encoding models. In a nutshell, ILA aims to enlarge intermediate-level perturbations in the direction of guiding examples, by maximizing projections on their mid-layer representations. As such, we first obtain the directional guides via applying gradient-based baseline attacks (*e.g.*, I-FGSM) maximizing $L_{\text{adversarial}}$, and ILA is then performed on the output of encoders.

It seems also possible to apply ILA using natural images from target classes as directional guides, and in this setting, no baseline attack is required. Yet, such easily obtained directional guides completely ignore local landscapes of the substitute models. By contrast, the proposed method locates vulnerable local regions of the model landscapes in the first place, making the directional guides more effective for ILA [19]. As demonstrated in the supplementary material, our method has obvious superiority.

# 4 Experiments

## 4.1 Setup

**General settings**   Same as many previous transfer-based work, we adopt the prediction accuracy of victim models on the generated adversarial examples as an evaluation metric. Lower accuracies indicate higher success rates. We focus on $\ell_\infty$ attacks, and tested our approach on two tasks: image classification and face verification. In the paper, we mostly utilize I-FGSM [26] for mounting attacks on the substitute models, in a way as described in Section 3.2. Some other baseline attacks including PGD [29] will be considered in Section A in the supplementary material. For image classification, we crafted adversarial examples based on benign ImageNet images [38], under the constraint of the maximum perturbation being no greater than 0.1 or 0.08, *i.e.*, $\epsilon = 0.1$ or 0.08. These images were chosen from the ImageNet validation set which contains $50\,000$ images in total. We randomly selected 5000 images from half of the ImageNet classes (*i.e.*, 500 classes and 10 images per class) to generate adversarial examples. For face verification, we first attacked open-source models on the LFW dataset [18], under a constraint of $\epsilon = 0.1$, and then tested with a commercial system held by `clarifai.com`. We randomly sampled 2110 images from over 400 identities from LFW, which are guaranteed not yet used in training all concerned victim models. Faces were aligned utilizing

MTCNN [58], and resized to $112 \times 112$ before mounting attacks. We rescaled the ImageNet and LFW images to a numerical range of $[0, 1]$, and clipped temporary results of all the concerned attacks at each iteration to ensure that the generated adversarial examples were valid images.

**Our approach**   As discussed, for crafting each adversarial example in the no-box setting, we would train a "substitute" model using the benign image to be perturbed, together with some other auxiliary images. We adopted the generator of CycleGAN [62] as a common auto-encoding architecture for our experiments. As mentioned, we should use no more than 20 images from two classes (*i.e.*, $n \leq 20$) to train each substitute model, on which a gradient-based baseline attack like I-FGSM can be readily performed. Specific number of training images for the two tasks will be later introduced in Section 4.2 and 4.3. We will discuss how the training scale affects the performance of our approach and training mechanisms in Section 4.4. Two unsupervised (*i.e.*, reconstruction from rotation and jigsaw) and one supervised (*i.e.*, prototypical reconstruction) training mechanisms have been introduced. Our approach equipped with these mechanisms will be called "***rotation***", "***jigsaw***", and "***prototypical***", respectively. We only considered four degrees: $0°, 90°, 180°$, and $270°$ for rotation, and for jigsaw, we uniformly cut the original images into four tiles and then shuffle, as illustrated in Figure 3. Models were all trained for at most $15\,000$ iterations using ADAM [24] with a fixed learning rate of $0.001$. Training could stop early if a performance plateau was reached on each tiny training set. Once a substitute model was ready, we first run a baseline attack (*e.g.*, I-FGSM) for 200 iterations and then run ILA for other 100 iterations, in a way as introduced in Section 3.2. $\lambda$ is simply set as 1 for all experiments. We let the optimization step-size of I-FGSM be $1/255$ for both ImageNet and LFW, following a bunch of prior work.

**Competitors**   We established two baselines by transferring adversarial examples from supervised ResNets and unsupervised auto-encoders conventionally trained on the same small-scale datasets as for our mechanisms, *i.e.*, each consists of no more than 20 images. The two baselines are codenamed "***naïve***[†]" and "***naïve***[‡]", respectively, in the paper[3]. Additionally, we further compared with adversarial examples transferred from models pre-trained on a large-scale and probably even the same dataset as training the victim models, codenamed "***Beyonder***" in the paper. I-FGSM was similarly adopted and we run it for 300 iterations for crafting each Beyonder adversarial example. The unsupervised auto-encoding naïve[‡] models were trained with the same policy as training our rotation/jigsaw/prototypical models. The supervised baseline naïve[†] models were trained with possible regularizations and data augmentations, though they assisted only to a limited extent. All our experiments were performed on one NVIDIA Tesla-V100 GPU using PyTorch [36] implementations.

**Victim models**   We tested very different victim models on ImageNet, including VGG-19 [42] with batch normalization [23], Inception v3 [45], ResNet-152 [15], DenseNet [17], SENet [16], wide ResNet (WRN) [56], PNASNet [28] and MobileNet v2 [39]. All the models are available in the Torchvision repository[4], except for PNASNet which is collected from Github[5]. Top-1 prediction accuracies of these models on the benign ImageNet images are summarized in Table 3 in the supplementary material. For face verification, we adopted FaceNet [40] [6] (whose backbone architecture is Inception-ResNet [44]) and Cosface [52] [7] (whose backbone architecture is a 64-layer ResNet-Like CNN), both trained on the CAISA-WebFace dataset [55].

## 4.2   Image Classification Results

We compared the proposed approach with Beyonder and the two baselines on ImageNet. A pre-trained ResNet-50 was collected as the Beyonder model. It shares the same training dataset with all the victim models, containing $1\,200\,000$ images from 1000 categories. By contrast, our approach and the two baselines (*i.e.*, naïve[†] and naïve[‡]) involve only 20 images to train each substitute model, *i.e.*, under the constraint of $n = 20$ for training.

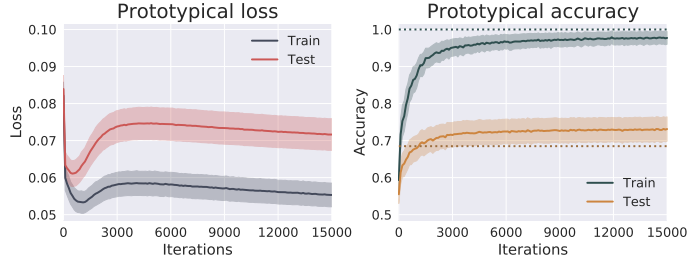

Figure 4: Our prototypical reconstruction mechanism leads to less over-fitting and higher *benign-set accuracy* of the substitute models in comparison with the conventional supervised models in Figure 1 and 2, using a small number of training images. The shaded areas indicate the amount of variance, and the dotted lines indicate final accuracies of the regularized VGG models in Figure 2.

For comparison, we report attack performance under $\epsilon = 0.1$ using supervised and unsupervised substitute models separately in Table 1. See Table 2 in the supplementary material for comparison under $\epsilon = 0.08$. For our approach with the prototypical mechanism, we tested both the single-decoder and multiple-decoder variants. It can be seen that the prototypical models with multiple decoders yield the most transferable adversarial examples overall, and, in particular, even outperform the Beyonder adversarial examples in attacking some ImageNet models, under $\epsilon = 0.1$. Visual explanations are given in Figure 6 in the supplementary material, and it can be seen that our adversarial examples divert the model attention from important image regions. Among unsupervised mechanisms, rotation seems to be the best, and it is worthy noting that our rotation and jigsaw mechanisms both outperform the unsupervised baseline (*i.e.*, naïve[‡]) obviously. Comparing the two baselines, naïve[‡] seems better in no-box transfer, benefiting from less over-fitting and better modeling of image patches.

We further depict the training curves of our multiple-decoder prototypical models in Figure 4. We cast the models into classifiers and illustrate accuracy curves as well, by making predictions based on the similarity scores as introduced in Section 3.1. Slightly different from the single-decoder models, here the similarity score for each class is obtained via averaging the Euclidean distance calculated with corresponding prototypes with different decoders. More specifically, an input $x$ is predicted into a class as

$$\hat{y} = \underset{y \in \{0,1\}}{\arg\min} \frac{1}{K} \sum_{k=0}^{K-1} \|\text{Dec}_k(\text{Enc}(x)) - x_k^{(y)}\|, \tag{5}$$

given the prototypes $\{x_k^{(y)}\}_{k=0}^{K-1}$ and the reconstruction result $\text{Dec}_k(\text{Enc}(x))$ on the $k$-the decoder, assuming there are in total $K$ decoders introduced in each prototypical model. We see that the multi-decoder prototypical auto-encoders achieve higher benign test accuracy than that of the supervised baseline, *i.e.*, naïve[†], and the gap between their training and test performance is also smaller. To be more specific, the prototypical models themselves show an average test-set accuracy of $73.12\%$ on benign images, while the supervised baseline using VGG and ResNet achieves an average prediction accuracy of $68.49\%$ and $63.65\%$, respectively, which are much lower. Moreover, training for more iterations may lead to even better performance with the prototypical models, since the training loss seems not fully converged in Figure 4, which is to be explored in future work.

Table 1: Compare the transferability of adversarial examples crafted on different models on ImageNet. The prediction accuracy on adversarial examples under $\epsilon = 0.1$ are shown (lower is better).

| Method | Sup. | VGG-19 [42] | Inception v3 [45] | ResNet [15] | DenseNet [17] | SENet [16] | WRN [56] | PNASNet [28] | MobileNet v2 [39] | Average |
|---|---|---|---|---|---|---|---|---|---|---|
| Naïve[‡] | ✗ | 45.92% | 63.94% | 60.64% | 56.48% | 65.54% | 58.80% | 73.14% | 37.76% | 57.78% |
| Jigsaw | ✗ | 31.54% | 50.28% | 46.24% | 42.38% | 59.06% | 51.24% | 62.32% | 25.24% | 46.04% |
| Rotation | ✗ | 31.14% | 48.14% | 47.40 % | 41.26% | 58.20% | 50.72% | 59.94% | 26.00% | 45.35% |
| Naïve[†] | ✓ | 76.20% | 80.86% | 83.76% | 78.94% | 87.00% | 84.16% | 86.96% | 72.44% | 81.29% |
| Prototypical | ✓ | 19.78% | 36.46% | 37.92% | 29.16% | 44.56% | 37.28% | 48.58% | 17.78% | 33.94% |
| Prototypical* | ✓ | 18.74% | 33.68% | 34.72% | 26.06% | 42.36% | 33.14% | 45.02% | 16.34% | **31.26%** |
| Beyonder | ✓ | 24.96% | 51.12% | 30.30% | 27.12% | 43.78% | 33.94% | 51.80% | 27.02% | 36.26% |

\* The prototypical models with multiple decoders. To be more specific, 20 decoders are introduced in each model.

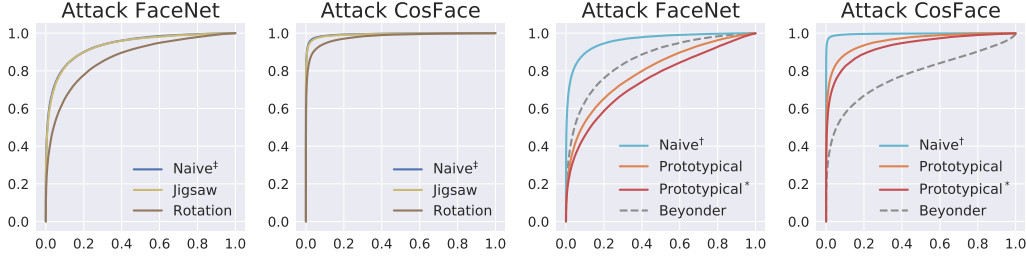

Figure 5: ROC curves of face verification on adversarial examples crafted on different substitute models. The left two sub-figures show *unsupervised* results and the right two show *supervised* results.

## 4.3 Face Verification Results

For the face verification task, we tested on the basis of LFW images [18]. Considering that a pair of faces are normally compared by calculating their cosine similarity in an embedding space, we slightly modified the adversarial loss in Eq. (4) to make it more suitable to the task. More specifically, we used the cosine similarity in lieu of the original Euclidean distance as:

$$L'_{\text{adversarial}} = -\log \frac{\exp\left(\lambda \frac{\langle \check{\text{Dec}}(\text{Enc}(x_i)), \check{\text{Dec}}(\text{Enc}(\tilde{x}_i)) \rangle}{\|\check{\text{Dec}}(\text{Enc}(x_i))\| \|\check{\text{Dec}}(\text{Enc}(\tilde{x}_i)))\|}\right)}{\exp\left(\lambda \frac{\langle \check{\text{Dec}}(\text{Enc}(x_i)), \check{\text{Dec}}(\text{Enc}(\tilde{x}_j)) \rangle}{\|\check{\text{Dec}}(\text{Enc}(x_i))\| \|\check{\text{Dec}}(\text{Enc}(\tilde{x}_j)))\|}\right)}, \tag{6}$$

in which $\check{\text{Dec}}(\text{Enc}(\cdot))$ denotes a mapping from the encoder outputs to an embedding space and we simply calculated at the output of ante-penultimate layers of the models. We used 10 images from two identities to train each substitute model in our no-box setting, *i.e.*, setting $n = 10$. We drew receiver operating characteristic (ROC) curves by testing the victim models on different sets of adversarial examples, in order to compare different approaches in a systematic manner. See Figure 5 for the ROC curves of the victim models in predicting adversarial examples crafted on different substitute models. It can be seen that the multiple-decoder prototypical models still achieve the best performance in attacking FaceNet, which is even better than that of Beyonder. They involve only 5 decoders here due to the lack of image prototypes. For unsupervised models, "rotation" seems the best training mechanism. Jigsaw models fail to learn, partially on account of the distortion of facial structures.

In addition to attacking the two famous verification models, we further demonstrate the performance of our approach on a commercial celebrity recognition system held by `clarifai.com`. We perform attacks using our generated adversarial examples under $\epsilon = 0.1$ and $0.08$, and the average prediction accuracies of the Clarifai system on these examples are merely $15.40\%$ and $33.67\%$, respectively. Beyonder with an Arcface [7] trained on millions of images gains $14.29\%$ and $21.85\%$, respectively.

## 4.4 Ablation Study

**Number of training images** The success of many deep learning machines rely on abundant training data. It is normally challenging to learn with a very small amount of data. We studied how the attack performance varied with the number of training images, *i.e.*, $n$. This experiment was performed on ImageNet. By varying $n$, we trained different sets of substitute models and mount attacks on them for attacking the no-box victim models. Detailed results are deferred to Figure 7 in the supplementary material. As expected, more data leads to more transferable adversarial examples on prototypical models. Somewhat surprisingly, our approach in general works reasonably well even with $n = 2$ and 10, *i.e.*, a no-box attack can possibly be performed by learning from 10 or even 2 images solely.

**Number of prototypical decoders** As shown in our main results, the proposed prototypical mechanism achieves the best performance in comparison with other mechanisms for training on the common small-scale data. Particularly, models with multiple decoders outperform their single-decoder counterparts. To study how the number of decoders affects the performance in more details, we compared our prototypical models with 1, 5, 10, and 20 decoders in an ImageNet experiment. We observed that the average prediction accuracy of the concerned victim models decreased from $33.94\%$ to $32.05\%$, further to $31.50\%$, and finally to $31.26\%$ in the settings. Detailed numerical results are also deferred to the supplementary material, due to the space limit of the main paper.

# 5 Conclusion

In this paper, we consider a realistic threat model for mounting adversarial attack, in which oracle queries and large-scale training are prohibited in addition to the internal information of victim models. We managed to train models using a limited amount of data, based on which transferable adversarial examples can be crafted. Different mechanisms for substitute model training in such a setting are specifically proposed. The best performing one, which is supervised, leads to discriminative models and adversarial examples that transfer well to a variety of advanced classifiers. The effectiveness of our approach is tested on image classification and face verification. Experimental results demonstrate that our approach outperforms possible baselines and it is sometimes even on par with transfer-based attacks from pre-trained models sharing the same large-scale training set with the victim models.

## Broader Impact

This paper pushes the boundary of adversarial attacks on machine learning models by introducing a novel attack under a stronger threat model, where the attacker has neither white-box nor black-box access to the victim model. We show that the performance of our attack is sometimes even on par with that of black-box attacks. This significantly raises the bar for defending machine learning models, because it will no longer be adequate to keep the models confidential (against white-box attacks), and to keep the training data confidential and further limit the number of queries (against black-box attacks). Our findings call the machine learning and security communities into action to create novel defenses and robust models. Robust models shall make AI applicable to a wider range of business sectors, particularly those that are safety-critical, and accessible to a broader population, particularly those who are pessimistic about the trustworthiness of AI.

Possible defenses to the proposed no-box attacks should be considered in future research. We found that adversarially trained models were not secure under these attacks, if the adversarial training was performed in a normal way (e.g., following the work of Madry et al.'s [29]). Specifically, our generated no-box adversarial examples led to a reasonably low accuracy (39.24%) on an adversarially trained ResNet, while the naive[†] baseline achieved 54.12%. A preliminary attempt of such a defense could be data augmentation using our adversarial examples, and it shall significantly improve the adversarial robustness to the proposed no-box attacks.

## Acknowledgment

This material is based upon work supported by the National Science Foundation under Grant No. 1801751. This research was partially sponsored by the Combat Capabilities Development Command Army Research Laboratory and was accomplished under Cooperative Agreement Number W911NF-13-2-0045 (ARL Cyber Security CRA). The views and conclusions contained in this document are those of the authors and should not be interpreted as representing the official policies, either expressed or implied, of the Combat Capabilities Development Command Army Research Laboratory or the U.S. Government. The U.S. Government is authorized to reproduce and distribute reprints for Government purposes not withstanding any copyright notation here on.

## Footnotes

[3]The naïve[‡] models share the same auto-encoding architecture with our rotation, jigsaw, and prototypical models, while the naïve[†] models are trained with regularizations and data augmentations as in Section 3.1.

[4]https://github.com/pytorch/vision/tree/master/torchvision/models

[5]https://github.com/Cadene/pretrained-models.pytorch

[6]https://github.com/timesler/facenet-pytorch

[7]https://github.com/MuggleWang/CosFace_pytorch

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
