[Supplementary Material]

# Practical No-box Adversarial Attacks against DNNs
## *Supplementary Material*

**Qizhang Li** *
ByteDance AI Lab
liqizhang@bytedance.com

**Yiwen Guo** †
ByteDance AI Lab
guoyiwen.ai@bytedance.com

**Hao Chen**
University of California, Davis
chen@ucdavis.edu

## A   More Experimental Results

$\ell_\infty$ **attacks under an $\epsilon$ of 0.08**   We compared different attacks under the constraint of $\epsilon = 0.08$ in Table 2. It can be seen that our approaches still outperforms the two baselines significantly. In general, the supervised mechanisms outperform the unsupervised ones. The performance of considered victim models on benign ImageNet images are reported in Table 3.

Table 2: Compare the transferability of adversarial examples crafted on different models on ImageNet. The prediction accuracy on adversarial examples under $\epsilon = 0.08$ are shown (lower is better).

| Method | Sup. | VGG-19 [7] | Inception v3 [8] | ResNet [1] | DenseNet [3] | SENet [2] | WRN [9] | PNASNet [4] | MobileNet v2 [5] | Average |
|---|---|---|---|---|---|---|---|---|---|---|
| Naïve[‡] | ✗ | 53.92% | 70.18% | 68.16% | 63.98% | 72.48% | 66.66% | 78.28% | 47.38% | 65.13% |
| Jigsaw | ✗ | 40.00% | 58.20% | 55.66% | 50.30% | 66.62% | 59.52% | 70.36% | 34.60% | 54.41% |
| Rotation | ✗ | 38.88% | 56.16% | 57.06% | 49.56% | 65.30% | 58.14% | 67.70% | 34.64% | 53.43% |
| Naïve[†] | ✓ | 76.64% | 81.24% | 83.98% | 79.54% | 87.14% | 84.30% | 87.12% | 73.16% | 81.64% |
| Prototypical | ✓ | 30.80% | 49.28% | 50.56% | 40.30% | 56.58% | 48.88% | 60.94% | 28.50% | 45.73% |
| Prototypical* | ✓ | 30.08% | 45.74% | 47.28% | 37.66% | 54.42% | 44.82% | 57.58% | 27.32% | 43.11% |
| Beyonder | ✓ | 27.70% | 53.58% | 33.74% | 30.58% | 46.70% | 37.26% | 54.92% | 29.42% | **39.24%** |

* The prototypical models with multiple decoders. To be more specific, 20 decoders are introduced in each model.

Table 3: Top-1 prediction accuracy of victim models on the randomly selected 5000 benign ImageNet images.

| Model | VGG-19 [7] | Inception v3 [8] | ResNet [1] | DenseNet [3] | SENet [2] | WRN [9] | PNASNet [4] | MobileNet v2 [5] | Average |
|---|---|---|---|---|---|---|---|---|---|
| Accuracy | 86.26% | 86.64% | 88.72% | 85.46% | 90.14% | 88.90% | 90.36% | 83.88% | 87.55% |

**Visualizations and explanations**   Here we visualize some adversarial examples and the model attention on the examples using Grad-CAM [6] in Figure 6. Grad-CAM provides interesting visual explanations of how our adversarial examples fool an advance victim model that is trained on millions of images. Here the results are obtained on the VGG-19 victim model. Obviously, our adversarial examples divert the model attention from important image regions, *e.g.*, from the distinctive body parts of the animals to irrelevant background regions. We also compare our generated no-box adversarial examples with the adversarial examples generated by Beyonder (which is basically just like in the white-box setting or a transfer-based black-box setting), and it can be seen that our no-box adversarial examples are intrinsically and perceptually very different from the Beyonder adversarial examples. Particularly, visual artifacts (somewhat like moiré patterns) may present in the no-box adversarial examples under $\epsilon$=0.1.

Figure 6: Visual explanation of how the Beyonder adversarial examples and our no-box adversarial examples fool the VGG-19 victim model. Grad-CAM is used.

**Number of training images**    We also tested our mechanisms in settings with even less training images. Though more data is more likely to lead to better performance for a supervised mechanism, we would like to know how the proposed mechanisms perform under more challenging circumstances with even less data. We summarize the performance of our rotation, jigsaw, prototypical (with single or multiple decoders) mechanisms on ImageNet in Figure 7. Lower prediction accuracy indicates better attack performance in the figure. It can be seen that all the proposed mechanisms perform reasonably well with no more than 20 images (*i.e.*, $n \leq 20$) on ImageNet. By further increasing $n$ to 40, the prototypical mechanism achieves even better performance in the sense of no-box transfer. Rotation and Jigsaw models seemingly works better with less training images, due to faster training convergence within the limited number of training iterations.

Figure 7: How the attack performance of our approach varies with the number of training images on ImageNet. Lower average accuracy indicate better performance in attacking the victim models.

**Number of prototypical decoders**    As mentioned in the main paper, we studied how the number of decoders would affect the attack success rate in our no-box setting. Table 4 summarizes the attack performance using our prototypical models equipped with 1, 5, 10, and 20 decoders in attacking the ImageNet models. Apparently, the more decoders get involved, the higher attack success rates can be achieved. However, it also takes longer to converge with more decoders, suggesting a trade-off between the attack success rate and training scale. It is somewhat unsurprising that multiple-decoder models outperform single-decoder models in mounting no-box attacks, since, as has been explained, richer supervision can be obtained from more decoders and more image anchors.

Table 4: How the number of prototypical decoders impact attack performance on ImageNet victim models. Results are obtained under $\ell_\infty$ attacks with $\epsilon = 0.1$. Lower is better.

| #decoders | VGG-19 [7] | Inception v3 [8] | ResNet [1] | DenseNet [3] | SENet [2] | WRN [9] | PNASNet [4] | MobileNet v2 [5] | Average |
|---|---|---|---|---|---|---|---|---|---|
| 1 | 19.78% | 36.46% | 37.92% | 29.16% | 44.56% | 37.28% | 48.58% | 17.78% | 33.94% |
| 5 | 19.48% | 34.32% | 35.90% | 26.44% | 42.70% | 34.72% | 46.12% | 17.37% | 32.13% |
| 10 | 19.16% | 34.18% | 35.00% | 25.94% | 42.14% | 33.16% | 45.22% | 17.18% | 31.50% |
| 20 | 18.74% | 33.68% | 34.72% | 26.06% | 42.36% | 33.14% | 45.02% | 16.34% | **31.26%** |

**Other baseline attacks** There exist other baseline attacks that can be used to craft adversarial examples on our substitute models. We tested different gradient-based baseline methods combined with ILA. As mentioned in Section 3.2 in the main paper, an image anchor from a different class (than that of the example to be perturbed) can be used as the directional guide. We tested such a strategy as well and denote it as "None+ILA". The obtained results are summarized in Table 5. Apparently, it does not perform as good as our introduced strategy applying I-FGSM with $L_{\text{adversarial}}$ first, which is denoted as "I-FGSM+ILA". As expected, PGD performs even better than I-FGSM, and it can be further explored. Note that PGD tested here incorporated randomness at each of its optimization iterations. The possibility of replacing ILA with other methods (e.g., TAP [10]) for improving the transferability was also considered. Specifically, our prototypical mechanism led to an average victim accuracy of 28.82% on ImageNet with TAP, under $\epsilon = 0.1$, which is remarkably superior to naive$^{\dagger}$ (77.39%) with TAP.

Table 5: Compare the transferability of different baseline attacks on the prototypical auto-encoding models on ImageNet, under $\epsilon = 0.1$. The prediction accuracy of the victim models on different sets of adversarial examples are shown (lower is better). PGD incorporates randomness in attacks, but we observed that the standard derivation of the attack performance among different runs are small (*e.g.*, it is only 0.06% for VGG-19, 0.12% for Inception v3, and 0.16% for ResNet), hence we omit it and only report the mean performance of "PGD+ILA" over 5 runs for clearer comparison in the table.

| Method | VGG-19 [7] | Inception v3 [8] | ResNet [1] | DenseNet [3] | SENet [2] | WRN [9] | PNASNet [4] | MobileNet v2 [5] | Average |
|---|---|---|---|---|---|---|---|---|---|
| None+ILA | 19.52% | 35.62% | 35.76% | 27.08% | 43.44% | 34.24% | 46.42% | 17.64% | 32.47% |
| I-FGSM+ILA | 18.74% | 33.68% | 34.72% | 26.06% | 42.36% | 33.14% | 45.02% | 16.34% | 31.26% |
| PGD+ILA | 18.02% | 32.06% | 33.64% | 23.62% | 40.78% | 31.88% | 43.64% | 14.94% | **29.82%** |

$l_2$ **attacks** In addition to the $l_\infty$ attacks, we also considered $\ell_2$ attacks. Specifically, by restricting the $\ell_2$ norm of the perturbations to be not greater than a common threshold, our prototypical mechanism led to a significantly lower average prediction accuracy (59.48%) of the victim models, in comparison to the supervised baseline (i.e., naïve$^{\dagger}$: 81.37%).

## Footnotes

*Work done during an internship at ByteDance