[Reviews · NeurIPS 2020]

Review 1

Summary and Contributions: This paper studies a very interesting and realistic no-box attack scenario: if only a few in-domain examples of the targeted model are available and nothing else, how should we conduct an effective attack? To this end, the authors first develop three methods to train auto-encoders to effectively capture the sample distributions and then attacks such well-trained auto-encoders to produce adversarial examples for fooling target systems. Extensive experiments on ImageNet and LFW are provided to demonstrate the effectiveness of the proposed method.

Strengths: (1) The whole paper is well written and easy to follow. (2) To my best knowledge, this is the first work that performs a systematic study on the no-box scenario, which is much more realistic than other attack settings (e.g., white-box attack). The authors also carefully demonstrate that crafting effective adversarial examples under the no-box setting is pretty challenging, e.g., naively train a substitute model on a few examples in a supervised manner cannot effectively break the target system. (3) The proposed attack framework is interesting and novel. First of all, since only a few examples are available, rather than naively training a discriminator, the author provides a much clever solution --- train an auto-encoder to effectively capture the sample distributions. Then the authors try to attack this well-trained auto-encoder to produce adversarial examples for fooling the target system. Specifically, three different ways to train auto-encoders are investigated, and the authors find that prototypical image reconstruction is the most effective way to train auto-encoder for improving the transferability of adversarial examples. (4) Extensive empirical results are provided to demonstrate the effectiveness of the proposed method. Moreover, the authors show that the proposed method can also effectively attack the commercial celebrity recognition system in the real world.

Weaknesses: Overall, I think this paper is a good one, and only with a few minor concerns: (1) In section 3.1, the authors claim “Training on different data from that of the victim models further leads to low attack success rates, implying that the internal statistics of images are not really captured.” Nonetheless, I do not find any experiments on the main paper to support this argument, do I miss anything? (2) By default, ILA [18] is applied in this work to boost transferability. I am a little bit curious about how other transferable attacks can help here (e.g., replace ILA with [48,56]). (3) I think it is interesting to provide some diagnoses on the adversarial perturbations generated by attacking auto-encoders. e.g., are these perturbations (visually) different from those generated by traditional methods? Also, if different ways are used to train auto-encoders (three ways studied in this paper), are the generated adversarial perturbations look different? (4) In the experiment, the perturbation size is 0.1. I am wondering if the added noises are noticeable under this setting? Can the authors provide any visualization of these generated adversarial examples? Also, if 0.1 indeed produces noticeable noises, will smaller perturbation size works for the proposed algorithm?

Correctness: Yes

Clarity: Yes

Relation to Prior Work: Yes

Reproducibility: Yes

Additional Feedback: --------------------------------------------Post-Rebuttal---------------------------------------------- I have read the rebuttal and other reviewers' comments. Overall, I do not have any major concerns on this paper and would like to accept it.


Review 2

Summary and Contributions: Thank you for your response, based on the reviewer discussion I encourage you to include some qualitative examples in the final version of the paper. This paper proposes the first practical adversarial attack on no-box models. While only having access to a small set of samples from the problem domain and no access to the model in neither a white-box nor a black-box fashion, substitute model training procedures for this challenging low-data setting are proposed based on auto-encoders. The generated representations allow for the generation of adversarial samples that are transferrable to the original classification problem. The method is demonstrated on both image classification and face verification datasets, clearly indicating that the generated adversarial samples indeed fool a variety of different models and lead to a significant deterioration in predictive accuracy.

Strengths: - The paper addresses the highly interesting no-box scenario where neither model internals nor model predictions can be accessed. Rather, only n<20 many samples from the joint data-label distribution are observed. The no-box assumption significantly expands the outreach of adversarial attacks to models which were previously believed to be safe under the white-box and black-box assumptions. - The paper is easy to follow for an adequately prepared reader. Prior work is sufficiently discussed and the two methodologies for training the auto-encoder-based architectures (dubbed "reconstruction from chaos" and "prototypical image reconstruction") as well as the procedure to generate the adversarial samples in these settings are clearly explained. - The experimental results show that adversarial samples crafted from the proposed approaches outperform naive solutions and even show comparable performance (in some instances) to transfer attacks from pre-trained supervised models sharing the same training set as the victim models, despite these method being unsupervised and only having access to a small amount of samples. This is a very interesting result and of high relevance to ML and security researchers.

Weaknesses: - Table 1 does not contain any sort of error bar its values, which makes it hard to quantify the robustness of these findings. While the authors mainly rely on the average of multiple different models to draw their final conclusions, it still would have been great to see whether any of the individual models are consistently more susceptible to the generated samples than others. The same is true for the facial verification results in Figure 5. - Unclear whether results hold beyond the L_inf norm. The approach's performance degrades sharply when comparing results from the main paper (eps=0.1) with results from the supplementary (eps=0.08)

Correctness: The claims made and the methods proposed in this work appear to be correct and valid. The experimental setup and evaluation also seems sound in general. However, as explained above, I missed some quantification of error (standard deviation or variance) for the numbers reported in Table 1.

Clarity: Overall, the paper is clearly written and easy to follow.

Relation to Prior Work: Related work in adversarial example generation, transferability of adversarial samples, and self-supervised learning is sufficiently discussed. While the no-box model has been introduced in prior work, no effective attacks on it have been demonstrated to date.

Reproducibility: Yes

Additional Feedback: The implementation details provided in the experimental results section seem sufficient to reproduce results. As noted throughout the paper, n<20 is used to constrain the number of total samples available from the underlying distribution. Another interesting setting, especially in a multi-class scenario, might be to not cap the number of total samples, but to have individual caps per class and perform an ablation study on that vector of constraints \bm{n} = (n_1, ... ,n_C) where C is the total number of classes and n_i is the cap of samples from class i. While this restriction can be looser in general, one could specifically model the cases when certain classes are not observed at all, which might also lead to some interesting results.


Review 3

Summary and Contributions: This paper studies the so called "no-box" adversarial attack in the tasks of image classification and face verification, where the attacker is given only access to a small set of data from the problem domain of the victim model. In particular, it introduces two unsupervised attack methods (rotation and jigsaw) and one supervised method (prototypical) and show that these methods achieve better results than the naive baseline methods and even better than adversarial attack transferred from a model trained with large amount of data in some cases. Further, the paper shows that their proposed attack can be used to attack a real-world commercial face verification system.

Strengths: - the proposed methods are well-motivated and novel. The unsupervised methods work under a even stronger threat model i.e.access to only a small set of unlabeled data of the same problem domain. - the experiments are comprehensive and the experiment results show the effectiveness of the proposed attack. In particular, this paper shows that their proposed method can be leveraged successfully to attack a real-world commercial face verification system. This experiment motivates the threat model very well. - the threat model is well motivated. The so-called no-box attack poses a more challenging scenario: keeping the model and the training data confidential are no longer enough. - code is provided.

Weaknesses: - No examples of images generated by the proposed attack are provided. The paper only evaluates performance of the proposed method under the setting of \epsilon = 0.08 and 0.1. At this level, the perturbation are likely to be perceivable by human.

Correctness: To the best of my knowledge, the claims and method are correct and the empirical methodology are in general correct. For the experiment, \epsilon=0.08 and 0.1 seem to be large enough for the attack to be perceivable. If they are indeed noticeable, the validity of the experiment setup will be hurt.

Clarity: Yes, it is well written.

Relation to Prior Work: Yes. However, I'd like to see a bit discussion about and comparison with universal adversarial perturbation in the related work section.

Reproducibility: Yes

Additional Feedback: Questions: - Could you provide some examples of the images generated by your attack? - How does your proposed method perform when \epsilon is smaller (say 0.03)? - In the Figure4 left image, why is the test prototypical loss lowest at first several hundred iterations but lower test accuracy? - Did you test how the adversarially trained models perform under this attack. Are the generated attacks similar to those generated from white-box / black-box methods? Suggestions: - I would like to see an ablation study on the parameter n. --------------------------------------------Post-Rebuttal---------------------------------------------- I have read all the reviews and the author's response. I am relatively satisfied with the author's response. The provided perturbed images (epsilon=0.1) in the response have some visual artifact. However, I tend to believe that the perturbed images under epsilon=0.03 should be more imperceptible and the proposed method also work seem to some extent under epsilon=0.03 in the provided additional results in the response. Thus, I increase my score to 7. For the final version, I would like to see more qualitative results under different epsilon to further verify the practical values of the perturbed images generated by the proposed method(s).


Review 4

Summary and Contributions: This work proposes an adversarial attack on deep neural network classifiers that does not require access to the model. The proposed attack does not fit into either the white-box or black-box attack categories - and is called a no-box attack. Specifically, the work proposes an attack that is generated by training auto-encoders using very few images (in the order of tens of examples) and transferring attacks generated using these auto-encoders to target models (that are unseen, and unqueried)

Strengths: 1. The research direction and general idea are very compelling. I believe no-box attacks are a very stimulating research direction that has not received much attention as of yet. 2. While the work is inspired by the idea of transferring attacks from surrogate models to target models; I believe the advances it presents regarding the amount of data that is necessary to train the "surrogate" and the very large differences between the proposed surrogate and the target model mean that a no-box attack can be effectively undertaken. Very interesting. I believe this specific idea has not been done before and that no-box attacks haven't really been demonstrated. This is to the best of my knowledge and would welcome any differing opinion by other reviewers. 3. It seems to me that the most "novel" attack version is the "prototypical" attack. But I actually believe that the contribution of the unsupervised versions "jigsaw" and "rotation" are incredibly interesting as well. I think the results are already surprising for those, and then the results for the "prototypical" attack show even better performance. 4. The paper presents results on two different classification tasks and two datasets. They present results on ImageNet, which is probably the best choice for object classification and on LFW, a solid choice for face verification. The experiments are very complete in my opinion (multiple datasets, multiple tasks, 3rd party black-box, multiple networks, multiple attacks, solid baselines). 5. The paper presents results on an online target model. The results shown are very satisfactory. 6. The amount of detail in the paper is very good. Also, the authors include the source code. I believe the algorithms should be reproducible given the amount of information provided. This is especially important in security research. I think this is the strongest paper in my batch of five papers this year.

Weaknesses: 1. I wouldn't say this is a weakness but it would be good to have some works on adversarial attacks on auto-encoders and image translation networks cited in the paper. [1,2] are important since they propose attacks on auto-encoders, and [3,4] are works on attacks on image translation networks for completeness. All of these works have the defining characteristic of attacking networks that have an image as an input and and image as an output and that the attacks are adapted from attacks such as FGSM, I-FGSM and PGD. This is related to the idea of the attack presented, since it is generated on an auto-encoder and transferred to a target model. None of these mentioned works diminish the novelty of the paper's ideas, they are just related. 2. Very small thing that made me chuckle, on line 214 I would change the informal "following a bunch of prior work". But that is just me. 3. The work could maybe try to find one or two papers that approach their scenario the best, and try to compare to these in the experiments section. I understand that other work does not approach this scenario exactly, but points of comparison are useful for readers at large and also to reviewers when making decisions. [1] Kos, Jernej, Ian Fischer, and Dawn Song. "Adversarial examples for generative models." 2018 ieee security and privacy workshops (spw). IEEE, 2018. [2] Pedro Tabacof, Julia Tavares, and Eduardo Valle. Adversarial Images for Variational Autoencoders. Adversarial Training Workshop, NIPS. 2016. [3] Ruiz, Nataniel, Sarah Adel Bargal and Stan Sclaroff. “Disrupting Deepfakes: Adversarial Attacks Against Conditional Image Translation Networks and Facial Manipulation Systems.” ArXiv abs/2003.01279 (2020) [4] Wang, Lin, Wonjune Cho, and Kuk-Jin Yoon. "Deceiving image-to-image translation networks for autonomous driving with adversarial perturbations." IEEE Robotics and Automation Letters 5.2 (2020): 1421-1428.

Correctness: Yes, it seems so, for the following reasons. 1. Hyperparameters and technical details are included. 2. Code is included. 3. Claims seem plausible. 4. Claims are supported by experiments. 5. Experiments are thorough. 6. Experimental setup seems correct.

Clarity: Yes very well written.

Relation to Prior Work: Good discussion of prior work. I had only one thing to say about it, and the addition of the papers that I mentioned (and others in that similar vein), could help this point.

Reproducibility: Yes

Additional Feedback: Very good paper. I really like the idea and I think it is well discussed and presented. The experiments were a pleasure to see, since they are concise, well presented and complete. I give it a score of 9.

[Author Response · NeurIPS 2020]

We are very grateful to the positive feedback and constructive comments. Minor comments will be addressed accordingly
and our responses to the major comments are provided as follows.

**Visualization and explanation:** We have provided some more visualization and explanation results in the below figure.
It can be seen that the no-box adversarial examples crafted on the auto-encoding models are indeed likely to be different
from the adversarial examples crafted on a pre-trained ResNet-50 (i.e., Beyonder) as in the white-box/black-box settings.
Particularly, visual artifacts (somewhat like moiré patterns) may present in the no-box adversarial examples under $\epsilon$=0.1.
Illustration results under $\epsilon = 0.08$ and $\epsilon = 0.03$ (where the perturbations are more imperceptible) will also be given, in
an updated version of the paper.

Figure 1: Visual explanation of the no-box adversarial examples and Beyonder examples. Grad-CAM is used.

**To Reviewer #1:**

1) Regarding our claim of "training on different data from that of the victim models further leads to low attack success
rates", we meant to say that the adversarial example crafted on models (with loss and accuracy shown in Figure 1 in the
paper) trained using a conventional supervised manner transferred poorly to the victim models. The experimental results
were given in Table 1 in the paper (reported as the "naive[†]" method). We will revise the related content for clarify.

2) **Q:** I am a little bit curious about how other transferable attacks can help here (e.g., replace ILA with [48,56]). **A:** We
tested with TAP [56] and observed that our mechanisms also worked well. Specifically, our prototypical mechanism led
to an average victim accuracy of 28.82% on ImageNet with TAP, which is remarkably superior to naive[†] (77.39%).

3) **Q:** Diagnoses on the adversarial perturbations generated by attacking auto-encoders? Is $\epsilon = 0.1$ noticeable? **A:** We
have provided more visualizations of the generated adversarial examples in this response letter. $\ell_\infty$ perturbations under
a constraint of $\epsilon$=0.1 may be perceptible on some images, therefore we have also reported attack performance under
$\epsilon$=0.08 in the supplementary material of the paper and $\epsilon$=0.03 in our response to Reviewer #3. Our method still works
significantly better than the concerned supervised and unsupervised baselines in these settings.

**To Reviewer #2:**

1) **Q:** Table 1 does not contain any sort of error bar. **A:** We here report the standard deviation of multiple training and
evaluations in the below table, and it can now be inferred which models are more susceptible to our no-box attacks.

Table 1: The standard deviation of the prediction accuracy on adversarial examples on ImageNet.

| Method | Sup. | VGG-19 | Inception v3 | ResNet | DenseNet | SENet | WRN | PNASNet | MobileNet | Average |
|---|---|---|---|---|---|---|---|---|---|---|
| Naïve[‡] | ✗ | 4.59% | 5.50% | 4.98% | 5.08% | 5.37% | 4.80% | 5.17% | 3.73% | 4.90% |
| Jigsaw | ✗ | 3.05% | 4.66% | 4.52% | 3.83% | 4.70% | 3.80% | 4.54% | 2.47% | 3.95% |
| Rotation | ✗ | 3.46% | 4.36% | 3.90% | 3.97% | 3.81% | 4.94% | 4.41% | 2.62% | 3.93% |
| Naïve[†] | ✓ | 5.66% | 5.61% | 5.32% | 6.59% | 4.42% | 5.22% | 4.55% | 6.55% | 5.49% |
| Prototypical | ✓ | 2.61% | 3.47% | 3.48% | 3.55% | 4.03% | 2.92% | 3.71% | 2.28% | 3.26% |
| Prototypical* | ✓ | 2.01% | 3.03% | 3.35% | 2.60% | 3.29% | 3.39% | 4.22% | 1.70% | 2.95% |

2) **Q:** Whether results hold beyond the $\ell_\infty$ norm? **A:** We further considered $\ell_2$ attacks. Specifically, by restricting the $\ell_2$
norm of the perturbations to be not greater than a common threshold, our prototypical mechanism led to a significantly
lower average prediction accuracy (59.48%) of the victim models, in comparison to the supervised baseline (81.37%).

**To Reviewer #3:**

1) **Q:** With smaller $\epsilon$ values, the success rates of no-box attacks diminishes. **A:** Indeed, under stricter constraints, the
performance of all the concerned attacks degrades. Yet, our prototypical mechanism still led to a much lower average
prediction accuracy (77.54%) of the victim models, than that of the supervised baseline (84.33%), under $\epsilon$=0.03.

2) **Q:** Lower test loss but lower test accuracy at first several hundred iterations. **A:** With very limited training data, the
supervised models became increasingly more confident during the first hundreds of training iterations, therefore several
incorrect predictions can lead to an increase of test loss (together with improvement in test accuracy though). Modeling
of the internal data distribution acted as a regularization and it actually alleviated this problem to some extent.

3) We followed the suggestion and tested on an adversarially trained ResNet provided by Madry et al.. The superiority
of our prototypical mechanism to its competitor held, leading to a lower victim accuracy (39.24% vs 54.12%).

4) We have performed an ablation study on $n$ in Section A in the supplementary material of the submission. It can be
seen that by further increasing $n$ to $40$, the prototypical mechanism achieved even better attack performance, while the
performance of rotation/jigsaw models degraded with larger $n$, probably on account of slower training convergence.

**To Reviewer #4:** Thanks for the very positive feedback! We will discuss the mentioned related work.

[Meta-Review · NeurIPS 2020]

The proposed approach to crafting attacks is original and the experiments are convincing. Please provide more qualitative examples for several magnitudes of perturbation in the final supplementary to meet the reviewers' requests. NOTE FROM PROGRAM CHAIRS: For the camera-ready version, please expand your broader impact statement to discuss the potential negative impacts of your work, as well as possible mitigations. In particular, please explain whether you believe there are risks to describing a new adversarial attack method without also proposing a defense.